# Exploring the Role of Large Language Models in Prompt Encoding for Diffusion Models

**Bingqi Ma**[1,*]       **Zhuofan Zong**[2,*]       **Guanglu Song**[1]

**Hongsheng Li**[2,3,4]       **Yu Liu**[1,✉]

[1] SenseTime Research       [2] CUHK MMLab

[3] Shanghai AI Laboratory       [4] CPII under InnoHK

{mabingqi, songguanglu}@sensetime.com

{zongzhuofan, liuyuisanai}@gmail.com       hsli@ee.cuhk.edu.hk

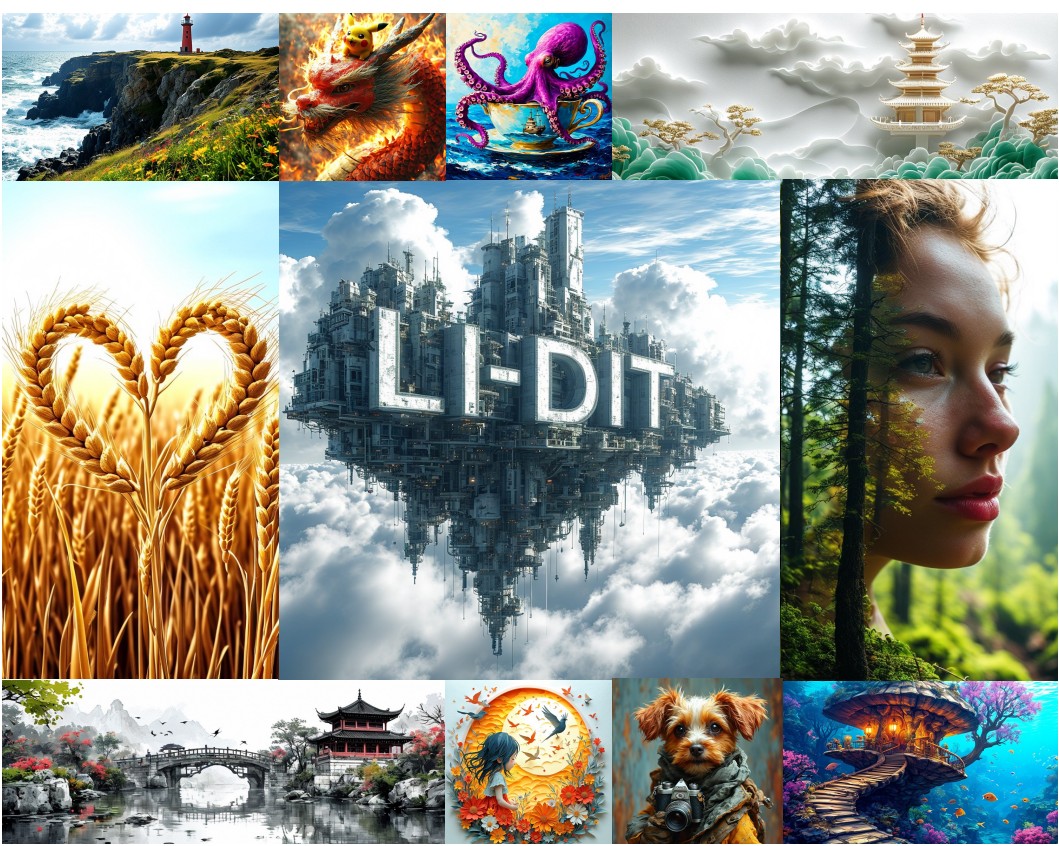

Figure 1: High-resolution (1024px) samples from our LI-DiT-10B, showcasing its capabilities in complex prompt comprehension, precise prompt following, and high image quality across various styles and resolutions. Please refer to the appendix for the prompts.

## Abstract

* Equal contribution. ✉ Corresponding author.

38th Conference on Neural Information Processing Systems (NeurIPS 2024).

Large language models (LLMs) based on decoder-only transformers have demonstrated superior text understanding capabilities compared to CLIP and T5-series models. However, the paradigm for utilizing current advanced LLMs in text-to-image diffusion models remains to be explored. We observed an unusual phenomenon: directly using a large language model as the prompt encoder significantly degrades the prompt-following ability in image generation. We identified two main obstacles behind this issue. One is the misalignment between the next token prediction training in LLM and the requirement for discriminative prompt features in diffusion models. The other is the intrinsic positional bias introduced by the decoder-only architecture. To deal with this issue, we propose a novel framework to fully harness the capabilities of LLMs. Through the carefully designed usage guidance, we effectively enhance the text representation capability for prompt encoding and eliminate its inherent positional bias. This allows us to integrate state-of-the-art LLMs into the text-to-image generation model flexibly. Furthermore, we also provide an effective manner to fuse multiple LLMs into our framework. Considering the excellent performance and scaling capabilities demonstrated by the transformer architecture, we further design an LLM-Infused Diffusion Transformer (LI-DiT) based on the framework. We conduct extensive experiments to validate LI-DiT across model size and data size. Benefiting from the inherent ability of the LLMs and our innovative designs, the prompt understanding performance of LI-DiT easily surpasses state-of-the-art open-source models as well as mainstream closed-source commercial models including Stable Diffusion 3, DALL-E 3, and Midjourney V6. The LLM-Infused Diffuser framework is also one of the core technologies powering SenseMirage, a highly advanced text-to-image model.

# 1 Introduction

The diffusion probabilistic models [1, 2, 3, 4, 5] have led to significant improvement in high-quality image synthesis. With the assistance of powerful prompt encoders such as the CLIP text encoder [6] and T5 series [7], DALL-E 3 [8] and Stable Diffusion 3 [9] greatly enhance the prompt understanding ability in text-to-image diffusion models. Encouraged by the success of GPT [10], a series of decoder-only large language models (LLMs) emerged and demonstrated superior text understanding capabilities compared to CLIP and T5 series models, e.g., LLaMA [11, 12]. However, methods for effectively leveraging these powerful LLMs in diffusion models remain to be explored [13, 14].

To better understand the inherent properties of LLMs in diffusion models, we first conduct experiments with the transformer-based diffusion model (DiT) [15] and perform evaluations on the T2I-CompBench [16] benchmark. Following the design in DiT and PixArt-$\alpha$[17], the text conditional information from the last layer of LLMs is injected into the diffusion transformer by cross-attention layers. As shown in Fig. 2, although LLaMA3-8B [1] exhibits much stronger language understanding ability [18], it still fails to catch up to the performance of the smaller model T5-XL on the image-to-text alignment benchmark. Meanwhile, the larger variant T5-XXL achieves a significant advantage over T5-XL. The powerful capabilities of LLMs in text comprehension and logical reasoning have not been demonstrated in such a scenario. Based on this anomaly, we aim to explore the role of LLMs in prompt encoding.

We start with analyzing the difference in optimization target and model architecture between T5-like encoder-decoder models and GPT-like decoder-only models. The masked language modeling optimization and the encoder-decoder architecture design endow the T5 encoder with an inherent ability for effective information comprehension. However, the optimization target of decoder-only LLMs focuses on predicting the next token with the highest probability based on training data distribution. As presented in Fig. 4, the pre-trained LLM provides a meaningless continuation to the given image prompt. It means that the LLM does not focus on the essential elements in the given image caption and the extracted text representation of LLM is not suitable for summarizing the semantic information of the given image, leading to a misalignment with the diffusion model's demand. Meanwhile, we find that LLMs generally cause errors or omissions in comprehending

---

[1]https://github.com/meta-llama/llama3

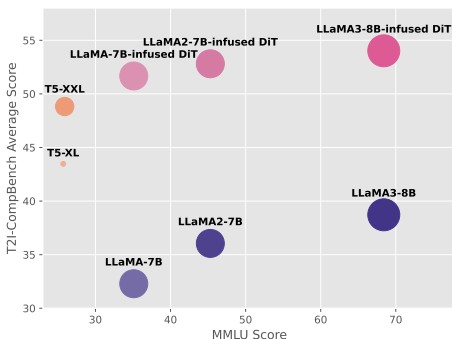
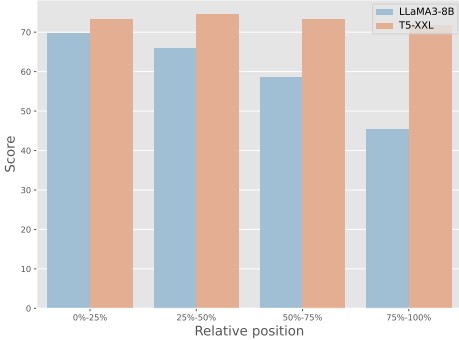

Figure 2: Comparisons of our model, LLaMA series, and T5 series on image generation and text understanding benchmarks.

Figure 3: Performance discrepancy between former and latter adj-noun compositions in LLaMA3-8B and T5-XXL.

objects or attributes mentioned in the latter part of the prompt. This observation is further validated through a quantitative evaluation. We attribute this issue to the causal attention mechanism of decoder-only LLMs. In the casual attention layer, each token can only attend to itself and other former tokens, while the information of the latter tokens cannot be captured. Such structured information imbalance challenges the diffusion model's ability to comprehend complex prompts. Therefore, the misalignment and positional bias significantly impede LLMs from being effective text encoders for diffusion models.

To address these issues, we propose a novel framework, LLM-infused Diffuser, to fully leverage powerful LLMs promoting diffusion models in text comprehension and following. First, we explicitly insert an instruction before the prompt to mitigate information misalignment. Based on the instruction-following ability of LLMs, we leverage human instruction to encourage language models focusing on concepts related to image generation, including objects, attributes, and spatial relations. Furthermore, we propose a linguistic token refiner to resolve the positional bias issue. Such designs facilitate effective global representation modeling via a bi-directional attention mechanism. Finally, the collaborative refiner merges and refines text representations from multiple LLMs to further boost text comprehension ability. These targeted designs provide an effective way to leverage the capabilities of LLMs in diffusion models.

Our LLM-infused Diffuser can be easily and flexibly incorporated into diffusion models. Considering the excellent performance and scaling capabilities of the transformer architecture [15, 9], we further design an LLM-infused Diffusion Transformer (LI-DiT). We conduct extensive experiments to validate LI-DiT across distinct model sizes and data sizes. Benefiting from the inherent ability of the LLMs and our innovative designs, the prompt understanding performance of LI-DiT easily surpasses state-of-the-art open-source models as well as mainstream closed-source commercial models including Stable Diffusion 3, DALL-E 3, and Midjourney V6. In Fig. 1, We present some randomly sampled cases generated by LI-DiT-10B.

## 2 Prompt Encoding with Language Models

As outlined in Sec. 1, we observe two discrepancies between decoder-only LLMs and encoder-decoder models: optimization objective and model architecture. Specifically, the decoder-only LLMs are typically optimized using the next token prediction task while the encoder-decoder models are trained with the masked language modeling task. Besides, the former tokens in a sequence cannot attend the latter tokens in decoder-only LLMs while every token in the sequence can attend each other in the encoder models. Based on the observations, we conduct elaborate experiments to investigate how such discrepancies affect the prompt encoding capacity of LLMs.

### 2.1 Exploring the Ability to Retain Prompt Information

During the pre-training of T5 models, the input sequences are formatted with masks, and the model learns from vast amounts of language data by predicting the masked content. In this process, the encoder is responsible for extracting information from all tokens in the current token sequence.

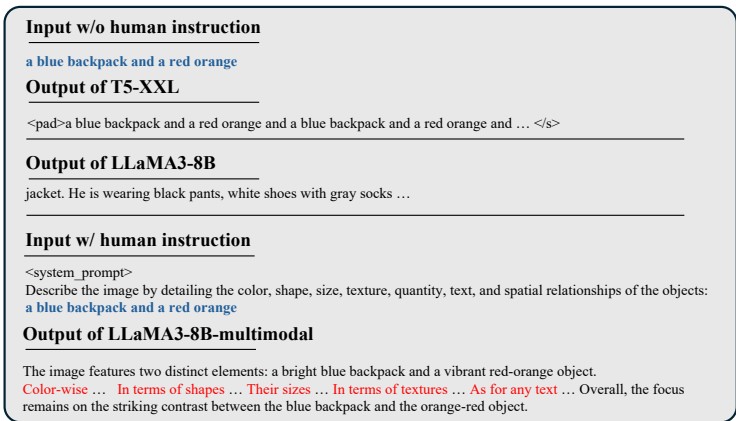

Figure 4: The output of language models when feeding a prompt. We can observe that pre-trained LLaMA3-8B provides an unrelated expansion, and T5-XXL repeats the input prompt. LLaMA3-8B with multi-modal fine-tuning can provide detailed information based on human instruction.

However, decoder-only language models focus more on predicting future information rather than representing the current text representation, which is misaligned with the diffusion model's usage. To better understand the characteristics of how language models encode prompts, we feed an image prompt into both LLaMA3-8B and T5-XXL to analyze their outputs. As shown in Fig. 4, the output of T5-XXL is the repeat of the input prompt while LLaMA3-8B generates an unrelated expansion. This phenomenon further validates our hypothesis. Therefore, even though LLMs possess stronger text understanding and reasoning capabilities, such limitation harms their capacity for encoding prompts.

## 2.2 Positional Bias of Decoder-only LLMs

We construct a benchmark to evaluate the image-text alignment of all adj-noun compositions at different positions in an image prompt. Following conventional text-to-image generation benchmarks [16, 13], we extract all adj-noun compositions and obtain their relative positions in each image prompt. These adj-noun compositions can be easily converted to questions. Then, we input the generated image and the question to a VQA model to obtain its alignment score. Please refer to the supplemental material for more details about constructing the test set. As shown in Fig. 3, we compute the average alignment score and the relative position within a prompt for each adj-noun composition. We can observe diffusion models with T5 encoders exhibit strong robustness to the position change, while models with decoder-only LLMs perform poorly in latter positions. Such inherent positional bias significantly harms the prompt encoding capacity of decoder-only LLMs.

## 3 LLM-infused Diffuser

### 3.1 Integrating LLMs and Diffusion Models

To bridge the gap between pre-training optimization and prompt encoding, we leverage the instruction-following capacity of LLM to encourage it to focus on image contents in the given caption. Furthermore, we also propose the refiner modules to mitigate the inherent positional bias of LLM text embeddings. By combining these designs, we develop a framework called LLM-infused Diffuser, which can flexibly infuse current state-of-the-art LLMs to unleash its strong text understanding capacity. As shown in Fig. 5, the pipeline of LLM-Infused Diffuser consists of four parts: (1) We insert the system prompt and instruction before the image prompt to encourage the LLM to focus on the image contents and highlight its attributes. (2) The image prompt with instructions can be encoded by multiple frozen LLMs separately. (3) Different linguistic token refiner modules are adopted to eliminate the positional bias of text embeddings from these LLMs. (4) With a collaborative refiner, text features from LLMs are collaboratively refined, resulting in more robust representations.

**Input Prompt.** Inspired by powerful instruction-following capabilities of LLMs [19], we aim to leverage such capabilities to force the LLM to attend to the crucial image contents in the prompt

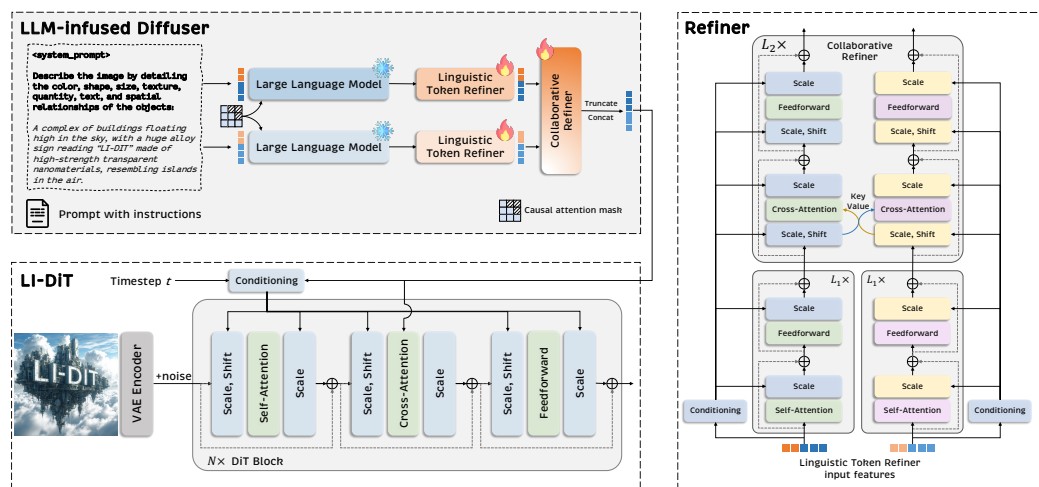

Figure 5: **The pipeline of LLM-infused diffuser.** First, the LLM-infused diffuser inserts an instruction to encourage LLMs to focus on image-related concepts. The linguistic token refiner eliminates the positional bias of LLM representations. Then the collaborative refiner further refines and mixes these embeddings and provides a more robust text representation. We only show 2 LLMs for simplicity.

and facilitate the alignment between the text representation and the text-to-image synthesis task. Specifically, we propose to insert the custom instruction before the conventional image description. Such instruction prompts the LLM to focus on critical image contents, such as object attributes and spatial relationships among objects in the image. In our experiments, we adopt a simple instruction: *Describe the image by detailing the color, shape, size, texture, quantity, text, and spatial relationships of the objects.* As shown in Fig. 4, the LLM tends to generate contents that are not related to the image context if we do not provide explicit instruction. When feeding the instruction and an image prompt to the LLM, it will follow the instruction to focus on the image-relevant concepts to detailedly describe the image and provide aligned representations based on the given prompt. The output embeddings of LLMs are further processed by subsequent refiner modules.

**Linguistic Token Refiner.** In the causal attention layer of LLM, only the previous tokens can be attended by the current token, thus it significantly hurts the global text representation modeling. For example, the last token in the text token sequence can only be attended by itself. To mitigate such positional bias of decoder-only LLMs, we insert a linguistic token refiner module to refine the biased output representations of each LLM. As shown in Fig. 5, each refiner module contains a stack of transformer blocks, which consists of a self-attention layer, a feed-forward layer (FFN), and an adaptive gating module. For the self-attention layer, we directly discard the causal mask of the LLM to perform full attention, which enables the representation of the latter token can be attended by former tokens. The output feature of each layer is controlled by adaptive gating networks, whose weights are initialized as zero for better training stability. To be specific, we first perform the average pooling to the LLM representation, then the pooled representation is merged with embeddings of the timestep $t$ via element-wise sum. The gating network takes such timestep-aware and context-aware representations as input to perform precise information injection. The final output representation of the refiner will be jointly fed into the collaborative refiner for enhancement.

**Collaborative Refiner.** To further improve text comprehension, we adopt multiple LLMs and linguistic token refiners for prompt encoding and collaboratively refine these representations through the proposed collaborative refiner. The representations from multiple linguistic token refiners are separately processed by multiple parallel branches and each block in a branch consists of a cross-attention and FFN layer. Besides, we use a modulation mechanism to condition each layer of collaborative refiner on the timestep and text context. This modulation takes the same input as the aforementioned gating network in the linguistic token refiner. The branches in this module are connected by multiple parallel cross-attention layers, where the text representations can be collaboratively refined. Specifically, the cross-attention layer takes the feature of the current branch as the query, and the features of other branches as the key and value to refine the current feature. Finally, We truncate the output token sequence, discard the instruction tokens, and mix both representations

by concatenation. This mixed and refined representation can be flexibly integrated into diffusion models to provide discriminative text conditional information.

## 3.2 LLM-infused Diffusion Transformer

Our proposed LLM-infused Diffuser can be flexibly integrated into current diffusion models. Considering the remarkable scaling capacity of diffusion transformers [15], we develop a diffusion model named LLM-infused Diffusion Transformer (LI-DiT).

Following the paradigm of DiT, LI-DiT takes the noisy representation from the latent space of a variational eutoencoder (VAE) as input and converts the spatial input into a sequence of tokens. Each transformer block of LI-DiT contains a self-attention layer, a cross-attention layer, an FFN layer, and the modulation module. The cross-attention layer can inject the text conditional information extracted by LLM-infused Diffuser into the token sequence. The modulation module receives the timestep embeddings and text representation to provide extra conditional information. Unlike the 2D positional embedding designs in previous works, we adopt a convolution-based position embedding. After the patchify layer in the diffusion transformer, we directly adopt a ResBlock [20] as the positional embedding module. The translation invariance of convolutional operators can effectively introduce positional information to the transformer operators. Therefore, LI-DiT can support arbitrary resolution image generation without requiring additional design modifications.

Large-scale transformer models usually suffer from unstable gradients and numerical precision, leading to divergent loss during training. To deal with the training instability issue, we incorporate several strategies adopted in large-scale vision or language model training. First, we introduce the QK-norm [21, 22] in both self-attention layers and cross-attention layers. The RMSNorm [23] layers will normalize the query and key tokens before the dot product computation of attention score. Such operation enables the numerical stability of attention scores and avoids unstable gradients from out-of-distribution values. Besides, considering the broader numerical representation range of bfloat16, we finally use the bfloat16 mixed precision training [24] strategy.

## 4 Comparing with Other Methods Adopting LLMs

Our LLM-infused diffuser has significant differences compared to the existing methods that utilize LLMs for prompt encoding. Apart from leveraging LLMs without specific design [25], current works can be classified into three categories. The first is that LLMs generate the image layout based on the prompt, and then the diffusion model completes the image based on this layout [26, 27, 28]. The second one is training an extra adapter to align LLM with frozen diffusion models like Stable Diffusion 1.5 [4] and Stable Diffusion XL [29] for better prompt comprehension capabilities [30, 31, 14, 13].

The contribution of the LLM-infused diffuser does not conflict with the layout approach. The layout methods are usually adopted as the controllable plugin in specific areas like visual composition and number-sensitive tasks. They need to be used in conjunction with a powerful diffusion model. However, the generation quality of each object in the layout still relies on the prompt understanding capability of the diffusion model. When generating a single object with a complex description, the layout approach essentially falls back to directly using the diffusion model for generation. Meanwhile, the layout can only provide the spatial relationship of objects but can not guide the generation of complex object relationships such as a boy sitting on the shoulder of a man, while the LLM-infused diffuser can easily deal with it. The adapter-based methods have not addressed the issues. LLM4GEN [31] also observed that the performance when adopting T5-XL can also easily outperform using larger 13B decoder-only LLMs. However, they did not provide any further analysis and directly used T5-XL as the final text encoder.

## 5 Experiments

### 5.1 Implementation Details

**Model Architecture.** Our experiments are conducted on the smaller model LI-DiT-1B by default. We adopt the LLaMA3-8B and Qwen1.5-7B [32] with multi-modal instruction fine-tuning [33] as the dual text encoders for both LI-DiT-1B and LI-DiT-10B. For the ablation study baseline, we only keep the LLaMA3-8B to reduce training costs. We adopt 2 blocks in the linguistic token refiner

Table 1: The performance of LI-DiT on T2I-CompBench, DPG-Bench and GenEval benchmark. We compare LI-DiT-1B with recent open-source academic works and compare LI-DiT-10B with mainstream closed-source commercial models. Experiments indicate the superior capabilities of LI-DiT on complex prompt understanding across the model size.

| Model | T2I-CompBench | | | | GenEval | | | | | | | DPG-Bench |
|---|---|---|---|---|---|---|---|---|---|---|---|---|
| | color | shape | texture | spatial | single | two | counting | colors | position | attribution | overall | average |
| SD v1.5 [4] | 37.50 | 37.24 | 41.59 | 12.04 | 0.97 | 0.38 | 0.35 | 0.76 | 0.04 | 0.06 | 0.43 | 63.18 |
| SD v2 [4] | 50.65 | 42.21 | 49.22 | 13.42 | 0.98 | 0.51 | 0.44 | 0.85 | 0.07 | 0.17 | 0.50 | 68.09 |
| SD XL [29] | 63.69 | 54.08 | 56.37 | 20.32 | 0.98 | 0.74 | 0.39 | 0.85 | 0.15 | 0.23 | 0.55 | 74.65 |
| SD3-1B [9] | - | - | - | - | 0.97 | 0.72 | 0.52 | 0.78 | 0.16 | 0.34 | 0.58 | - |
| DALL-E 2 [34] | 57.50 | 54.64 | 63.74 | 12.83 | - | - | - | - | - | - | - | - |
| PixArt-α [17] | 68.86 | 55.82 | 70.44 | 20.82 | 0.98 | 0.50 | 0.44 | 0.80 | 0.08 | 0.07 | 0.48 | 71.11 |
| LI-DiT-1B | 74.08 | 59.34 | 69.59 | 27.57 | 0.98 | 0.69 | 0.48 | 0.86 | 0.22 | 0.37 | 0.60 | 81.65 |
| DALL-E 3 [8] | 81.10 | 67.50 | **80.70** | - | 0.96 | 0.47 | 0.47 | 0.83 | 0.43 | 0.45 | 0.67 | 83.50 |
| SD3-8B [9] | - | - | - | - | **0.99** | **0.94** | **0.72** | 0.89 | 0.33 | 0.60 | 0.74 | - |
| LI-DiT-10B | **83.78** | **68.03** | 78.50 | **39.69** | **0.99** | 0.91 | 0.65 | **0.91** | **0.47** | **0.64** | **0.76** | **84.60** |

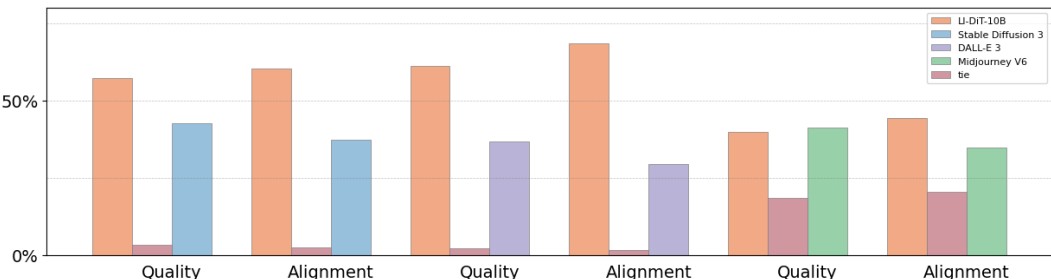

Figure 6: Human evaluation performance. Our LI-DiT-10B surpasses other open-source and close-source leading text-to-image generators on both quality and alignment. We can observe that LI-DiT-10B surpasses Stable Diffusion 3 and Dall-E 3 on both quality and alignment. Compared with the most popular Midjourney V6, LI-DiT-10B demonstrates leading capabilities in image-text alignment with similar image-text quality performance.

and 1 block in the collaborative refiner. In our experiments, we take the text embedding from the third-to-last transformer block as the output of each LLM. For the detailed architecture of LI-DiT-1B and LI-DiT-10B, please refer to the supplementary materials.

**Training Data.** All the exploration and ablation experiments are trained on the ImageNet dataset [35] and a subset of the CC12M dataset [36]. We assign the text prompt of "*a photo of {class}*" to each sample of ImageNet and randomly select 1.3M image-text pairs from CC12M. Following previous works [9], we mix the original captions and synthetic captions generated by CogVLM [37]. When we compare LI-DiT with other leading counterparts, we employ a large-scale training dataset with billion-level image-text pairs, including LAION-5B [38] and other internal datasets containing both English and Chinese, which enables LI-DiT with bilingual comprehension capabilities. Following stable diffusion [4], we remove the image-text pair from LAION when its aesthetic scorer is lower than 4.7. Low-resolution images and low-quality prompts including URLs and tags are also removed. Specifically, we only sample a subset of 30M image-text pairs from this large-scale dataset to train LI-DiT-1B and use all the billion level pairs to train the LI-DiT-10B.

**Training Details.** Following the paradigm of latent diffusion models (LDM) [4], we leverage a VAE encoder [39] to project the image representation into the latent space. We train a VAE with $8\times$ downsample rate and 16 channels for better image generation [9]. We do not use any data augmentation strategies. Following the multi-scale training in RAPHEL[40], we group the images based on their aspect ratio. Only images with similar aspect ratios will construct a batch. For the ablation experiments conducted on 3M image-text pairs, we train the models with a batch size of 256 and a learning rate of 1e-4 for 300k iterations at 256 resolution. For the training of LI-DiT-1B, we increase the batch size to 2048 and iterations to 500k. When training LI-DiT-10B, the batch size is 4096, and the iteration number is over 1M. We directly employ a resolution of 512 during training, and then fine-tune it to 1024 resolution with high-quality data to further improve the aesthetic quality.

**Evaluation Metrics.** For the quantitative evaluation, we mainly consider the T2I-CompBench [16], DPG-Bench [13], and GenEval benchmark [41]. We also introduce human evaluations for better

Table 2: Component-wise ablation.

| instruction | token | collaborative | T2I-avg | DPG-avg |
|---|---|---|---|---|
| | | | 38.72 | 66.15 |
| ✓ | | | 48.41 | 73.45 |
| | ✓ | | 54.02 | 77.08 |
| ✓ | ✓ | | 56.79 | 78.62 |
| ✓ | ✓ | ✓ | **60.31** | **80.25** |

Table 3: Effects of causal mask.

| LLM | token refiner | full attn | T2I-avg | DPG-avg |
|---|---|---|---|---|
| T5 | | ✓ | 48.82 | 73.56 |
| T5 | ✓ | ✓ | 49.52 | 74.63 |
| Qwen1.5 | | | 38.11 | 65.61 |
| Qwen1.5 | ✓ | ✓ | **53.81** | **76.49** |
| LLaMA3 | | | 38.72 | 66.15 |
| LLaMA3 | ✓ | | 45.84 | 71.01 |
| LLaMA3 | ✓ | ✓ | **54.02** | **77.08** |

Table 4: Effect of instruction.

| multi-modal | instruction | T2I-avg | DPG-avg |
|---|---|---|---|
| | | 38.72 | 66.15 |
| | ✓ | 38.47 | 65.84 |
| ✓ | | 44.22 | 71.81 |
| ✓ | ✓ | **48.41** | **73.45** |

Table 5: Token refiner design.

| N | gating | T2I-avg | DPG-avg |
|---|---|---|---|
| 1 | ✓ | 48.25 | 73.83 |
| 2 | ✓ | 54.02 | 77.08 |
| 3 | ✓ | **55.13** | **77.65** |
| 2 | | 53.47 | 76.68 |

Table 6: Fusion design.

| setting | T2I-avg | DPG-avg |
|---|---|---|
| LLaMA | 56.79 | 78.62 |
| Qwen | 56.13 | 78.49 |
| concat | 58.32 | 79.04 |
| refiner | **60.31** | **80.25** |

comprehension of the artistic and aesthetic qualities. Note that the "T2I-avg" in ablation studies refers to the average score of T2I-CompBench attribute metrics.

## 5.2 Performance Comparisons

**Quantitative Evaluations.** In the quantitative evaluation, we focus on the alignment between generated images and the input prompts. As shown in Tab. 1, we choose T2I-CompBench, DPG-Bench, and GenEval benchmark to evaluate the generation capability of LI-DiT-1B and LI-DiT-10B. The T2I-CompBench and the GenEval benchmark are composed of short prompts, focusing on the compositional evaluation. The DPG-Bench is built with complex dense prompts. Compared with open-source academic works like SDXL and PixArt-α, LI-DiT-1B outperforms them over all benchmarks by a large margin. We also compare LI-DiT-10B with DALL-E 3 and Stable Diffusion 3 (8B), two mainstream closed-source commercial models. The significant improvement further validates the effectiveness of our LLM-Fused Diffuser.

**Human Evaluations.** The quantitative evaluation metrics can not directly measure the artistic and aesthetic qualities. Following previous works, we conduct the human evaluation to convincingly compare LI-DiT-10B with Stable Diffusion 3, DALL-E 3, and Midjourney V6. Our evaluation dataset contains 200 prompts with diverse styles and scenarios. The image from LI-DiT-10B and the image from a competitor will construct an evaluation pair. The human evaluator will compare the image pair from the perspective of image quality and image-text alignment. The result in Fig. 6 indicates that LI-DiT-10B can surpass DALLE-3 and Stable Diffusion 3 in both image-text alignment and image quality. Compared with the most popular commercial model Midjourney V6, LI-DiT-10B demonstrates leading capabilities in image-text alignment with similar image-text quality performance. In Fig. 7, we show some randomly sampled cases to make a clear comparison.

## 5.3 Ablation Study

**Componet-wise ablation study.** As shown in Tab. 2, we conduct the component-wise ablation study. We adopt DiT with pre-trained LLaMA3-8B as the baseline setting. First, we observe consistent performance gains after introducing the instruction to the input prompt or incorporating the linguistic token refiner to the baseline. When leveraging both designs, the image-text alignment performances on two benchmarks continue to improve. Besides, we introduce an extra powerful LLM, Qwen1.5-7B with multi-modal fine-tuning to verify the effectiveness of the collaborative refiner. The LLM fusion strategy further enhances the prompt comprehension ability of the diffusion model. These results clearly validate the effectiveness of each proposed component.

**Effect of causal mask.** We investigate the effect of the causal mask on prompt encoding in this experiment. As presented in Tab. 3, inserting a linguistic token refiner with full attention after the LLM significantly improves the performance. However, this refiner fails to increase the performance of the T5 encoder with bi-directional attention. If we introduce the causal mask of LLM to the refiner, severe performance degradation occurs in both LLaMA3-8B and Qwen1.5-7B. These results demonstrate the causal mask is a core factor that harms the prompt encoding capacity of the LLM and our proposed refiner can eliminate such positional bias.

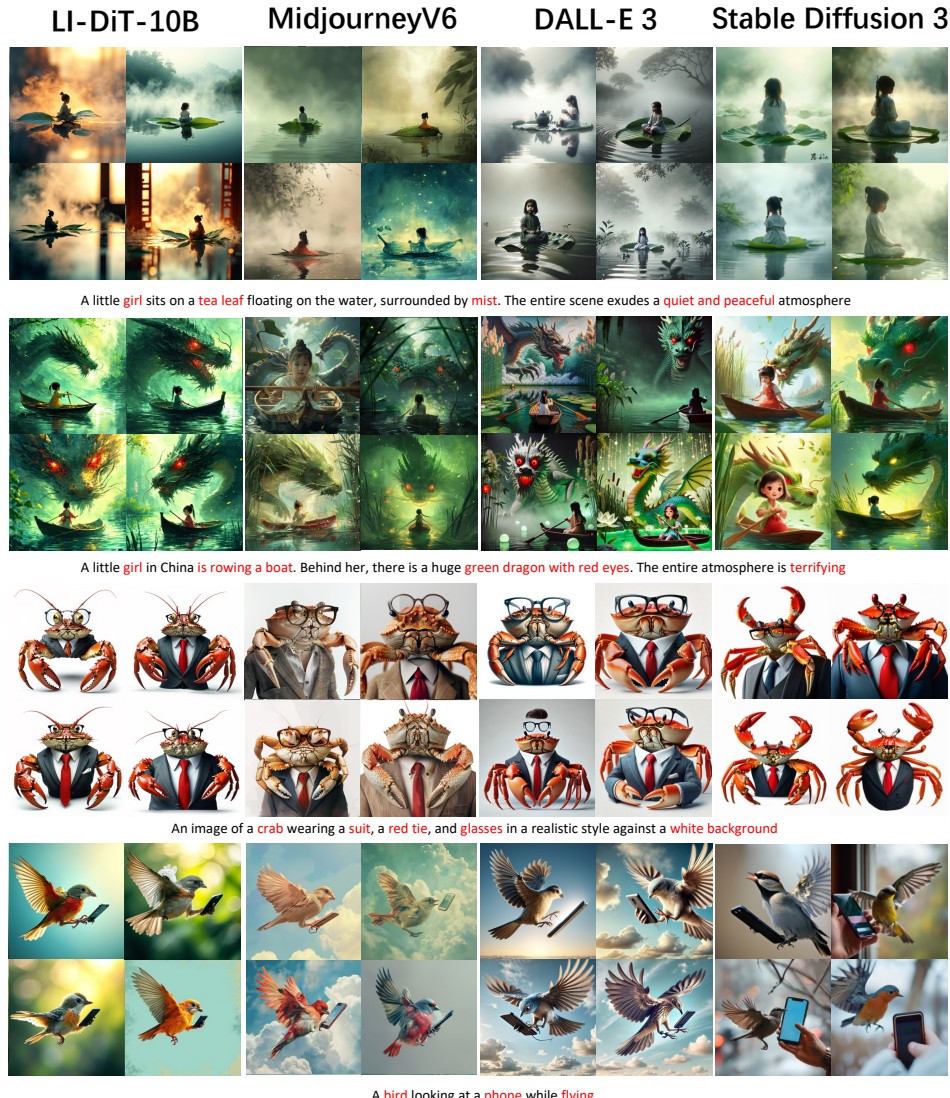

| LI-DiT-10B | MidjourneyV6 | DALL-E 3 | Stable Diffusion 3 |

A little girl sits on a tea leaf floating on the water, surrounded by mist. The entire scene exudes a quiet and peaceful atmosphere

A little girl in China is rowing a boat. Behind her, there is a huge green dragon with red eyes. The entire atmosphere is terrifying

An image of a crab wearing a suit, a red tie, and glasses in a realistic style against a white background

A bird looking at a phone while flying

Figure 7: Comparisions with Midjourney V6, DALL-E 3 and Stable Diffusion 3. The prompts are randomly sampled from our human evaluation benchmark.

**Effect of instruction.** To verify the effectiveness of the instruction, we conduct an ablation in Tab. 4. First, we find the prompt instruction fails to bring gains for the model that employs a base LLaMA3-8B without instruction fine-tuning. If we institute the base model for a multi-modal instruction fine-tuned variant, the alignment scores can be significantly increased. Thanks to the strong instruction-following capacity brought by instruction fine-tuning, inserting the instruction can further boost performance. This result demonstrates the multi-modal instruction fine-tuning data helps the LLM better describe an image and highlight key elements within the image. Besides, the instruction is able to encourage the LLM to attend to the image contents in the given prompt.

**Linguistic token refiner design.** As shown in Tab. 5, we conduct experiments on the design of linguistic token refiner. First, we compare our model with other variants with different numbers of blocks in the refiner. We observe consistent performance gains when the number of blocks in the refiner increases. However, such gain is not significant when there are 2 blocks in the linguistic token refiner. Therefore, we employ 2 blocks in the token refiner to achieve the best trade-off between complexity and performance. Besides, we also ablate the effect of the gating network in the refiner. When we remove the gating network, the performances on both benchmarks decrease. This indicates that the conditional information of time and text context contributes to better image-text alignment.

**Effect of collaborative refiner.** As shown in Tab. 6, we observe the model with a simple fusion technique can outperform the other counterparts with a single LLM. Besides, the collaborative refiner can further boost the performance based on this concatenation fusion. Such a result indicates that an effective representation fusion method can further enhance the capabilities of LLMs.

## 6 Related Work

**Diffusion models.** The denoising diffusion probabilistic model (DDPM) [1] provides an effective manner to generate high-quality images. To train diffusion models on limited computational resources while retaining their quality and flexibility, the latent diffusion models (LDMs) [4] project the images into the latent space of pre-trained autoencoders [39]. A time-conditional UNet [42] is applied to denoise from the noisy latent input. Please refer to the supplementary materials for detailed information about the optimization process. The transformer architecture has achieved remarkable success in various tasks. Dit [15] is the pioneering work in adopting transformer architecture in diffusion models. Transformer models exhibit excellent scaling properties [22], which support the training of large-scale diffusion models. Recent advanced models [17, 43, 9, 25, 44, 45, 46, 47, 48] in image generation and video generation mainly consider the transformer architecture as the backbone. Apart from the DDPM paradigm, Stable Diffusion 3 [9] and Lumina-T2X [25] leverage the flow matching [49] strategy to optimize diffusion models.

**Text encoder for diffusion models.** The CLIP text encoder [6] is popular among various text-to-image generation models [34, 29, 4]. Under the image-text contrastive optimization, the CLIP text encoder can map prompts into a unified image-text space, providing valuable information for conditional image generation. Meanwhile, utilizing CLIP text encoders with larger parameters and more extensive training data [50, 51] has significantly enhanced the diffusion model's ability to comprehend prompts [29, 9]. Imagen [5] observes that large language models like T5 [7] pre-trained on text-only corpora are surprisingly effective at encoding text for image synthesis. Recent works [17, 43, 52, 8, 9] usually adopt the T5 series as the prompt encoding model. Considering the excellent text comprehension capabilities of decoder-only LLMs [11, 12, 53, 32, 54, 55, 56], some works [25, 14, 13, 57] try to introduce LLMs into the designed framework. However, systematic comparative analysis on T5 models and LLMs is still missing. LLMs with instruction fine-tuning like Vicuna [58] exhibit powerful instruction following capabilities. The multi-modal instruction fine-tuning [59, 60, 61, 37, 62, 63, 64, 65, 66, 67, 68, 69, 70, 71] further enables LLMs to understand visual information. These models have the potential to serve as reliable text encoders.

## 7 Conclusion

In this paper, we explore the role of LLMs in prompt encoding for diffusion models based on the poor performance in the text-to-image generation task when adopting a decoder-only LLM to encode prompts. Through experiments and analysis, we identified the core factors limiting decoder-only LLMs as effective text encoders for diffusion models are the misalignment between next token prediction training and the requirement for discriminative prompt features in diffusion models, and the intrinsic positional bias introduced by the decoder-only architecture. To deal with the issues, we propose a novel framework to fully harness the capabilities of LLMs. We further design an LLM-Infused Diffusion Transformer (LI-DiT) based on the framework. LI-DiT surpasses state-of-the-art open-source models as well as mainstream closed-source commercial models including Stable Diffusion 3, DALLE-3, and Midjourney V6.

## 8 Limitation and Potential Negative Societal Impact

Due to the limited computation resources, we conduct experiments on LLMs with 7B parameters. In future work, we will further validate the effectiveness of LLM-infused Diffusion in larger LLMs with 13B or 70B parameters. The potential negative social impact is that images may contain misleading or false information. We will conduct extensive efforts in data processing to deal with the issue.

**Acknowledgments**   The work was supported by the National Key R&D Program of China under Grant 2021ZD0201300.

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

# A Appendix

## A.1 Denoising Diffusion Probabilistic Model

The optimization target of DDPMs can be defined by maximizing the log-likelihood of the training data. Given the data distribution $q(\mathbf{x}_0)$, the forward diffusion process is defined as:

$$q(\mathbf{x}_{1:T}|\mathbf{x}_0) = \prod_{t=1}^{T} q(\mathbf{x}_t|\mathbf{x}_{t-1}), \tag{1}$$

where

$$q(\mathbf{x}_t|\mathbf{x}_{t-1}) = \mathcal{N}(\mathbf{x}_t; \sqrt{\alpha_t}\mathbf{x}_{t-1}, (1-\alpha_t)\mathbf{I}). \tag{2}$$

Here, $\alpha_t$ is the noise schedule parameter. The reverse diffusion process is the key part of training, where a parameterized model $p_\theta$ is learned to approximate the reverse process of the data:

$$p_\theta(\mathbf{x}_{0:T}) = p(\mathbf{x}_T) \prod_{t=1}^{T} p_\theta(\mathbf{x}_{t-1}|\mathbf{x}_t), \tag{3}$$

where

$$p_\theta(\mathbf{x}_{t-1}|\mathbf{x}_t) = \mathcal{N}(\mathbf{x}_{t-1}; \mu_\theta(\mathbf{x}_t, t), \sigma_t^2\mathbf{I}). \tag{4}$$

The optimization objective of DDPM is to minimize the Variational Lower Bound (VLB), thus maximizing the log-likelihood of the model. The optimization objective can be expressed as:

$$\mathcal{L}_{\text{VLB}} = \mathbb{E}_{q(\mathbf{x}_{0:T})}\left[\log\frac{q(\mathbf{x}_{1:T}|\mathbf{x}_0)}{p_\theta(\mathbf{x}_{0:T})}\right]. \tag{5}$$

By decomposing and rewriting, we get the following form:

$$\mathcal{L}_{\text{VLB}} = \mathbb{E}_{q(\mathbf{x}_{0:T})}\left[\sum_{t=1}^{T} D_{KL}(q(\mathbf{x}_t|\mathbf{x}_{t-1})\|p_\theta(\mathbf{x}_t|\mathbf{x}_{t+1})) - \log p_\theta(\mathbf{x}_0|\mathbf{x}_1)\right]. \tag{6}$$

Simplified and restated as the loss at each timestep:

$$\mathcal{L}_t = \mathbb{E}_{q(\mathbf{x}_0, \mathbf{x}_t)}\left[\frac{1}{2\sigma_t^2}\|\epsilon - \epsilon_\theta(\mathbf{x}_t, t)\|^2\right], \tag{7}$$

where $\epsilon$ is the noise. These formulas describe the optimization objective of the DDPM model. By minimizing this loss function, the model can effectively learn the data distribution and progressively denoise during the generation process, producing high-quality data samples.

## A.2 Detailed information of LI-DiT-1B and LI-DiT-10B

In Tab. 7, we provide the detailed architecture and training information of LI-DiT-1B and LI-DiT-10B.

Table 7: The detailed architecture of LI-DiT-1B and LI-DiT-10B

| model | depth | hidden size | head number | patch size | input resolution | batch size | iter | training data |
|-------|-------|-------------|-------------|------------|------------------|------------|---------|---------------|
| LI-DiT-1B | 28 | 1152 | 16 | 2 | 256 | 2048 | 500k | 30M |
| LI-DiT-10B | 48 | 2816 | 44 | 2 | 512/1024 | 4096 | over 1M | over 1B |

## A.3 Evaluation Benchmark Construction for Positional Bias

In this chapter, we primarily discuss the construction of the positional bias evaluation benchmark. We used the attributes and nouns provided by T2I-CompBench to construct 1,000 prompts, each containing up to 8 nouns or attributes. For each prompt, the diffusion model will generate 4 images. We divided each prompt into segments (adj-noun composition). Following the design in T2I-CompBench, we used BLIP [63] to score the alignment of segments at different positions. When testing performance, we first calculate the average score within each prompt segment, then compute the overall average score for all prompts to obtain the model's accuracy within that segment. We provide a few samples as follows:

- *A green bench, a red car, a blue bowl, and a pink apple.*

- *A black banana, a yellow bird, a blue dog, and a brown horse.*

- *A metallic car, a wooden desk, a rubber band, and a metallic knife.*

- *A plastic cutlery, a fabric shirt, a fluffy pillow, and leather gloves.*

- *A big elephant, a small flea, a diamond pendant, and a round watch.*

- *A round bagel, a rectangular knife block, a tall lighthouse, and a short buoy.*

## A.4  Prompts in Fig. 1

We provide the prompts adopted to generate images in Fig. 1. The prompts are arranged from left to right, top to bottom.

- *A dramatic coastal cliff scene with waves crashing against the rocks below. The cliffside is covered in green grass and wildflowers, and a lighthouse stands tall on the edge, overlooking the vast ocean. The sky is partly cloudy, with the sun peeking through.*

- *A Chinese dragon with a Pikachu on its head, featuring fire effects.*

- *A surreal painting features a giant octopus with vivid purple tentacles emerging from a large teacup, while a miniature ship floats on the surface. The whimsical seascape blends ocean waves with fantastical elements, creating a dreamlike atmosphere. Vibrant colors and playful light reflections enhance the scene, inspired by fantasy art, and rendered in high definition for an immersive experience.*

- *A digital art piece using C4D modeling blends Wang Ximeng's landscape art with jade carving and multi-layered paper-cutting. Featuring the Yellow Crane Tower, white jade-carved clouds, and sculpted buildings, it incorporates crystal and glass for a digital feel. Predominantly white, the artwork highlights exquisite craftsmanship and lighting.*

- *A golden wheat field with two ears of wheat forming a heart shape in the center of the image. The background is the sky under the midday sun.*

- *A complex of buildings floating high in the sky, with a huge alloy sign reading "LI-DIT" made of high-strength transparent nanomaterials, resembling islands in the air.*

- *A photo portrait of a handsome woman and beautiful forest, double exposure*

- *An ink wash painting with abundant brushstrokes and a heavy sense of history. It features ancient Chinese gardens with gray walls, black tiles, pavilions, boats, flowers, and trees. A stone bridge spans the water, with intricately arranged rockeries, evoking the serene atmosphere of a Jiangnan water town.*

- *A multi-dimensional paper-cutting art piece features a little girl beneath a glowing moon, surrounded by flying birds and flowers. The watercolor illustration uses warm colors on a light background, with exquisite details and 3D rendering. Pastel hues and soft light create a dreamy, delicate atmosphere, resulting in a high-quality, visually captivating design.*

- *A handsome little dog carrying a camera on its shoulder.*

- *A massive treehouse built within a giant conch shell, intricate wooden bridges, and lanterns adorning the shell's spiral. The background is a vibrant coral reef with colorful marine life. Soft, underwater lighting with shimmering reflections.*

## A.5 High-quality Images Showcases.

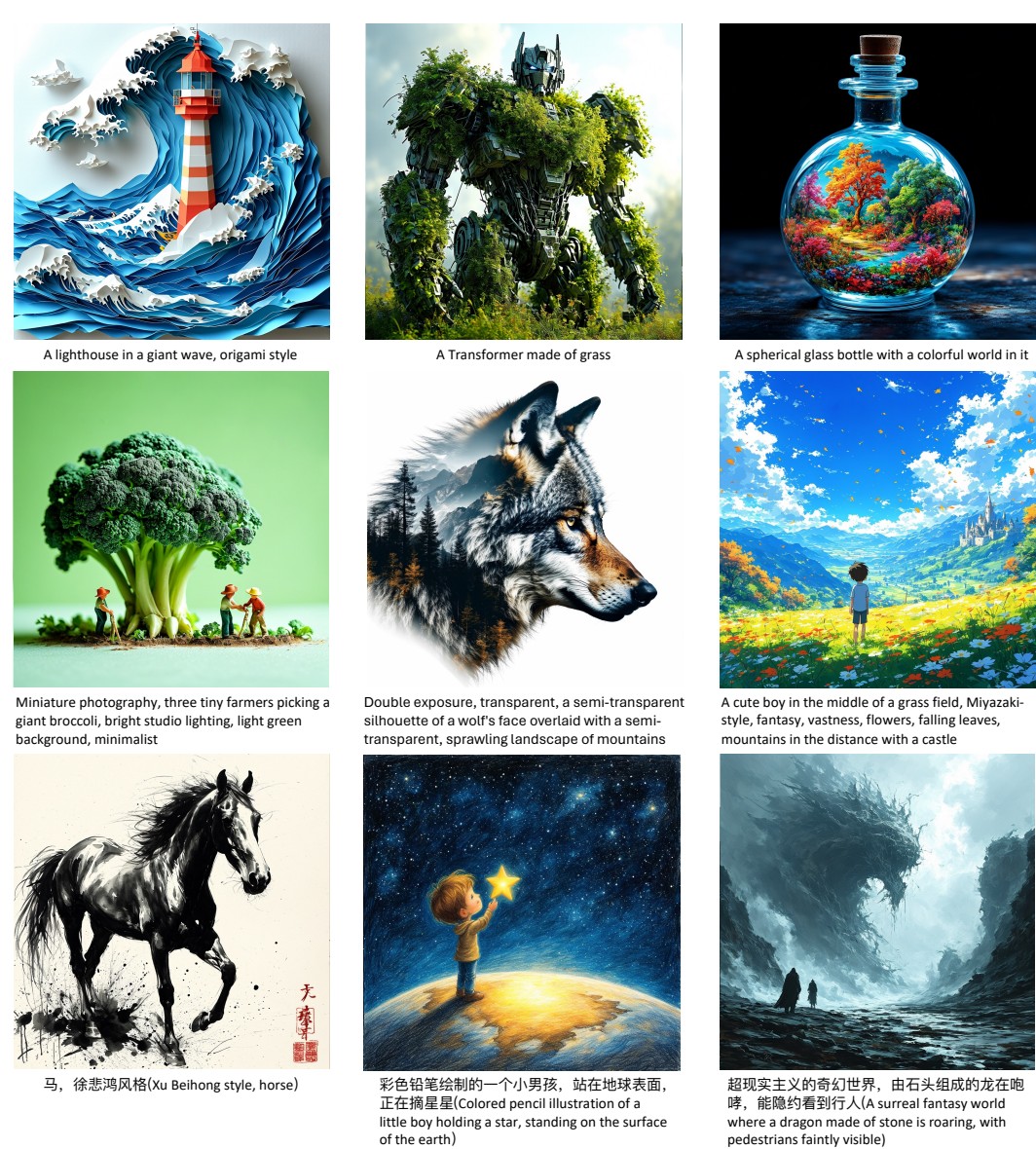

Figure 8: LI-DiT-10B exhibits an astonishing ability to understand bilingual prompts, accurately generating images even with complex descriptions and combinations of objects.

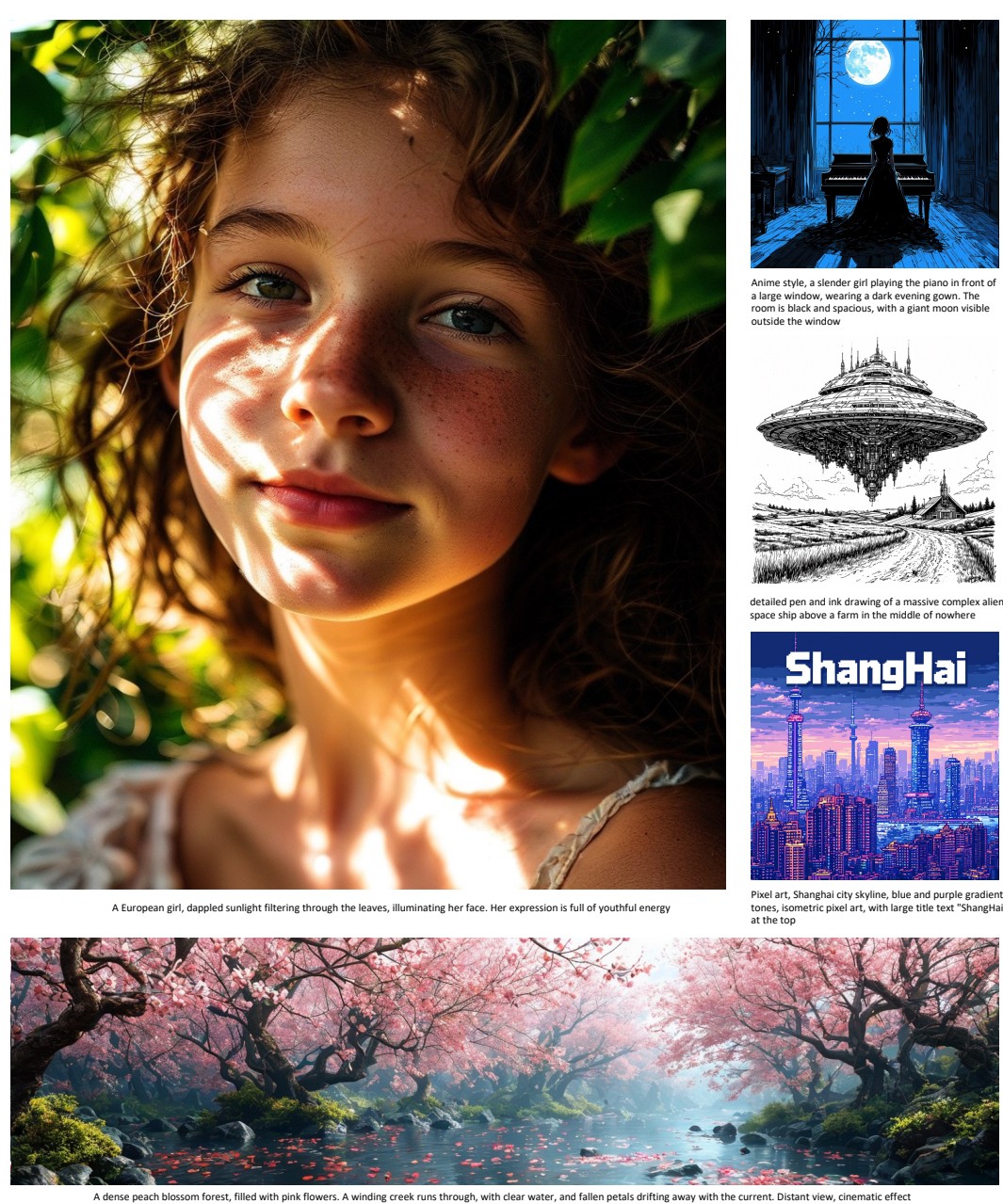

A European girl, dappled sunlight filtering through the leaves, illuminating her face. Her expression is full of youthful energy

Anime style, a slender girl playing the piano in front of a large window, wearing a dark evening gown. The room is black and spacious, with a giant moon visible outside the window

detailed pen and ink drawing of a massive complex alien space ship above a farm in the middle of nowhere

Pixel art, Shanghai city skyline, blue and purple gradient tones, isometric pixel art, with large title text "ShangHai" at the top

A dense peach blossom forest, filled with pink flowers. A winding creek runs through, with clear water, and fallen petals drifting away with the current. Distant view, cinematic effect

Figure 9: LI-DiT-10B exhibits an astonishing ability to understand prompts, accurately generating images even with complex descriptions and combinations of objects.

## A.6 Comparison with Other Models.

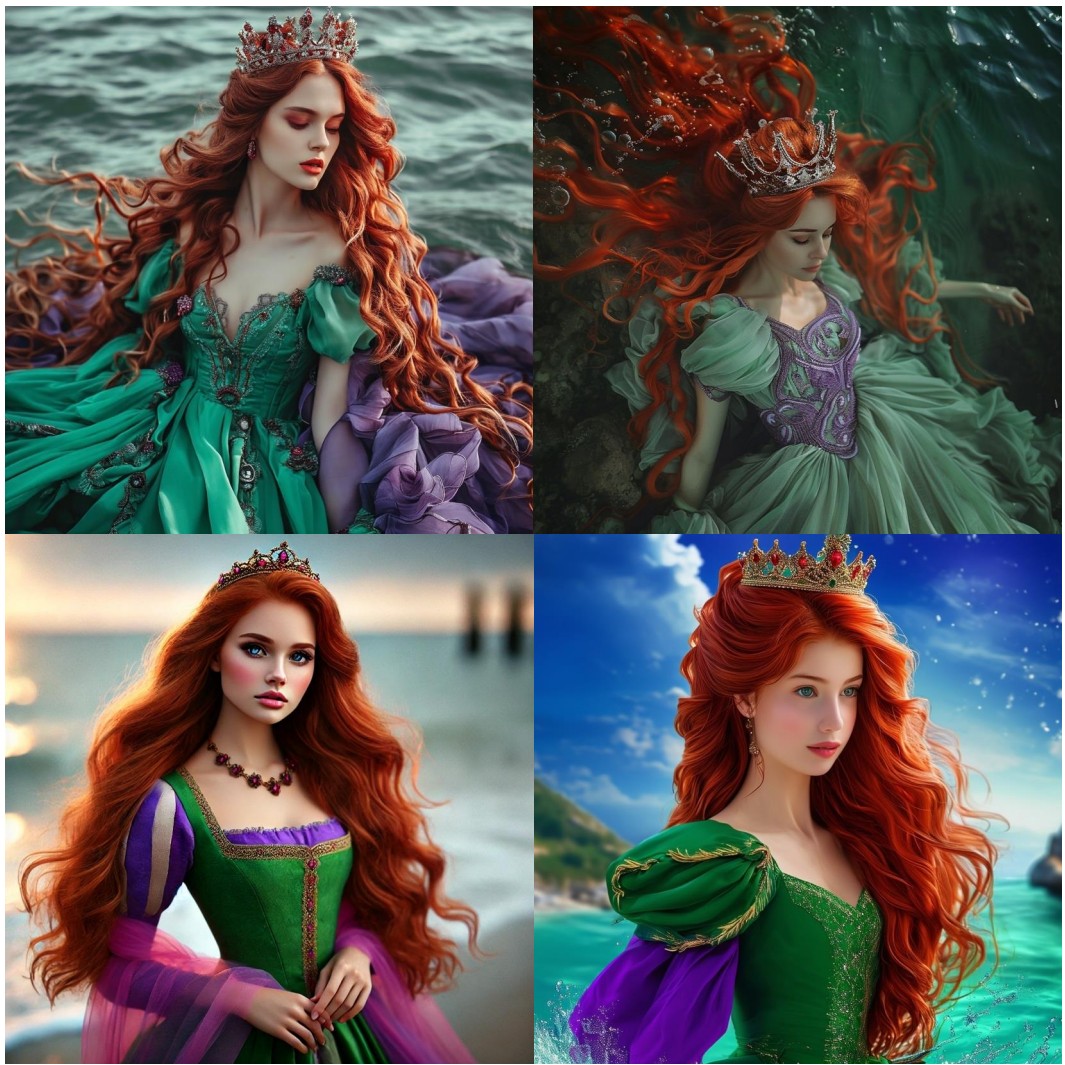

Real photography, a princess wearing a green dress, purple clothes, her hair is very long and red, very beautiful, wearing a crown on her head, living in the sea.

Figure 10: Comparisions with Midjourney V6, DALL-E 3 and Stable Diffusion 3. The prompts are randomly sampled from our human evaluation benchmark. The images are presented in the order of LI-DiT-10B, Midjourney V6, DALL-E 3, and Stable Diffusion 3.

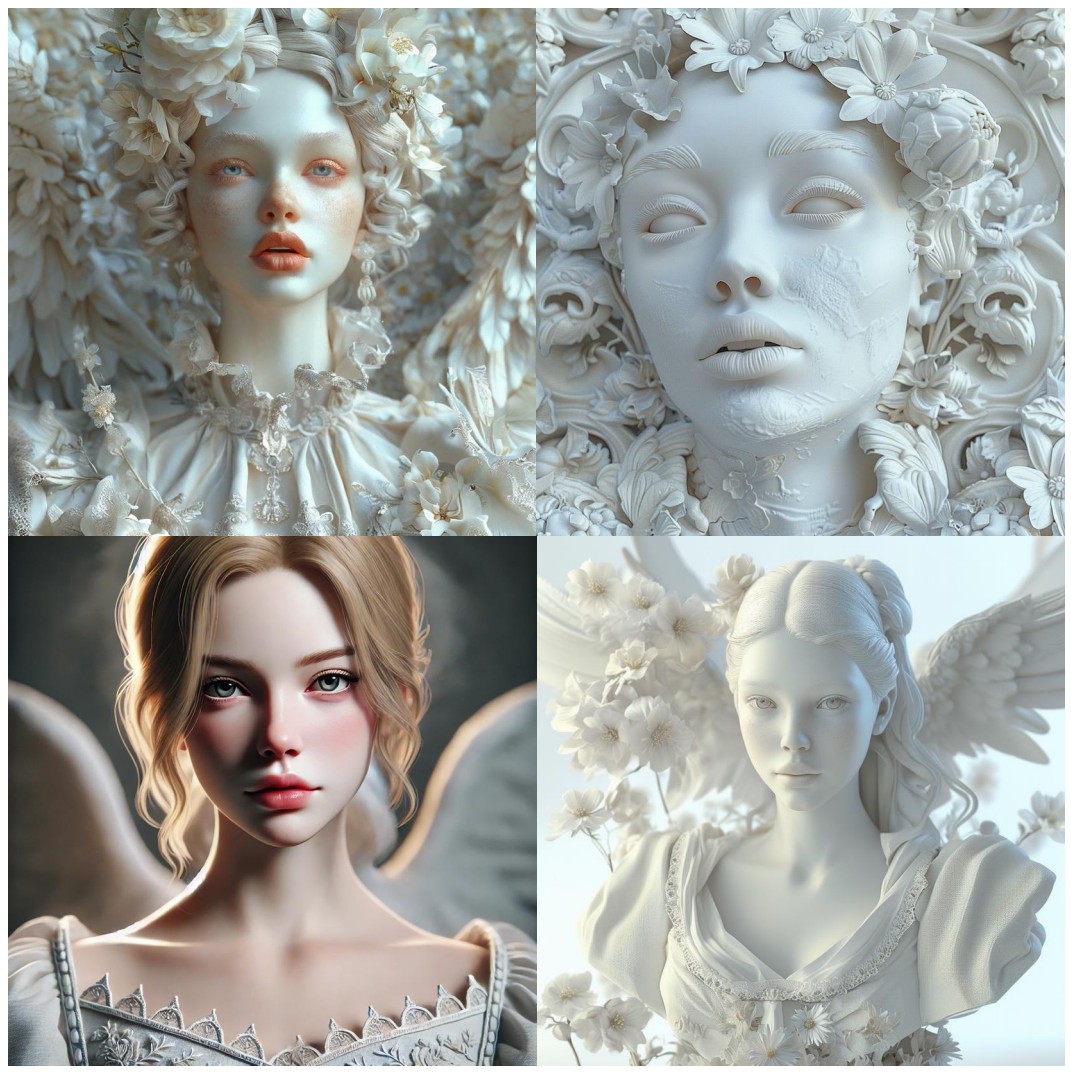

3D, Octane render，bust of a white，skinned woman with light eyes, thick lips, thin nose, fine white fabric dress, with angels and flowers, Renaissance style

Figure 11: Comparisions with Midjourney V6, DALL-E 3 and Stable Diffusion 3. The prompts are randomly sampled from our human evaluation benchmark. The images are presented in the order of LI-DiT-10B, Midjourney V6, DALL-E 3, and Stable Diffusion 3.

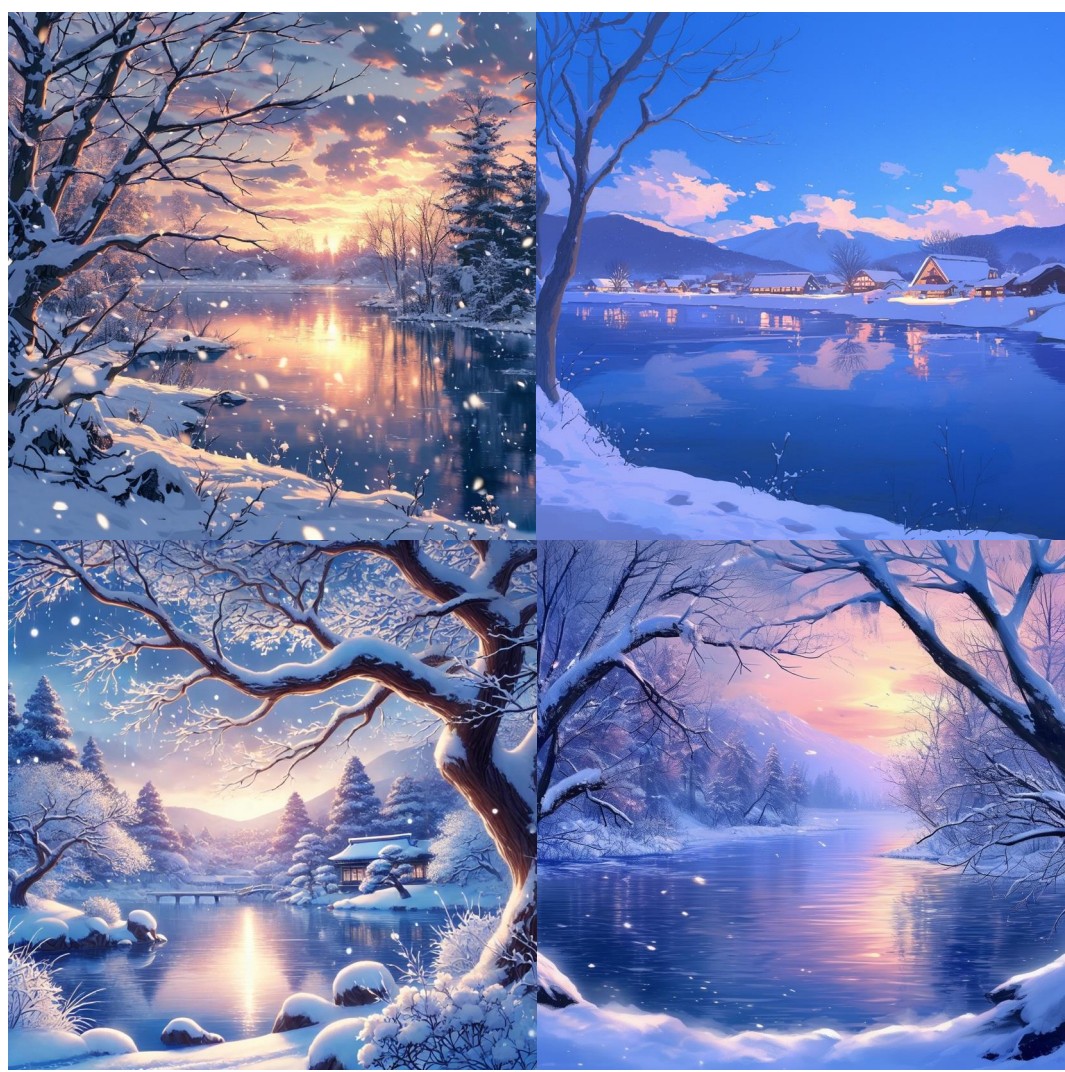

Anime, peaceful snow scenery, peaceful lake surface, snow covered branches, winter wonderland in the twilight, picturesque scenery, Thomas Kinkade style, magical and peaceful, high quality.

Figure 12: Comparisions with Midjourney V6, DALL-E 3 and Stable Diffusion 3. The prompts are randomly sampled from our human evaluation benchmark. The images are presented in the order of LI-DiT-10B, Midjourney V6, DALL-E 3, and Stable Diffusion 3.

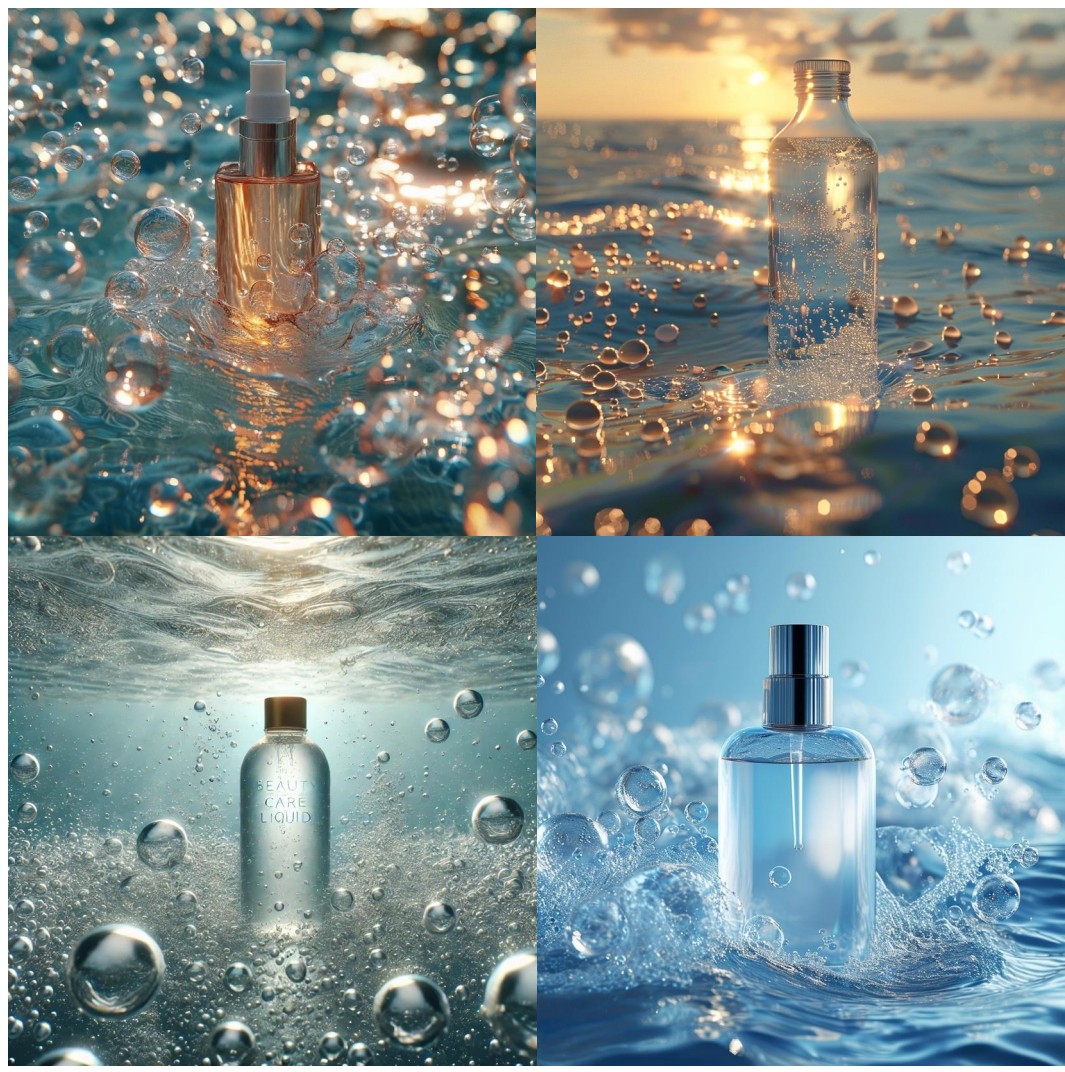

A bottle of beauty care liquid sank into the sea and is surrounded by bubbles. There are too many bubbles. Soft light is refracted through the sea water. The large water ripple network makes the picture beautiful, high resolution, fine detail, front view, 8K

Figure 13: Comparisions with Midjourney V6, DALL-E 3 and Stable Diffusion 3. The prompts are randomly sampled from our human evaluation benchmark. The images are presented in the order of LI-DiT-10B, Midjourney V6, DALL-E 3, and Stable Diffusion 3.

