# OpenReview forum: "Exploring the Role of Large Language Models in Prompt Encoding for Diffusion Models"
_NeurIPS.cc/2024/Conference — NeurIPS 2024 poster_

### Official Review · Reviewer_nE1N · 2024-07-09

**Soundness:** 3
**Presentation:** 3
**Contribution:** 3
**Rating:** 6
**Confidence:** 3

**Summary:**

The paper addresses the challenges of using large language models (LLMs) as prompt encoders in text-to-image diffusion models. It identifies two primary issues: the misalignment between LLM training objectives and the requirements of discriminative prompt features in diffusion models, and the positional bias introduced by the decoder-only architecture of LLMs. To tackle these issues, the authors propose a novel framework called LLM-infused Diffuser, which leverages human instructions and linguistic token refiners to enhance text representation capabilities. They design an LLM-Infused Diffusion Transformer (LI-DiT) and demonstrate its superior performance over state-of-the-art models in both open-source and commercial systems.

**Strengths:**

* Introduction of LLM-infused Diffuser to facilitate the integration of LLMs into diffusion models to boost generation performance.
* Experiments validate the effectiveness of this proposed framework when compared with both SOTA open- and closed-source baselines.

**Weaknesses:**

* Scalability: While the paper demonstrates the superior performance of LI-DiT, the scalability of the approach to other diffusion models is not fully discussed and validated.
* Lack of training and inference costs: Despite the superior generation quality, the paper does not provide detailed information on the training and inference costs associated with the proposed model.

**Questions:**

* Could the authors provide information on both the training and inference costs associated with the proposed method, such as GPU memory consumption, training time, and inference time?

* The paper mentions that the LLM-infused Diffuser can be easily and flexibly integrated into diffusion models. Does this imply that once trained, the LLM-infused Diffuser can seamlessly integrate into existing diffusion models without further fine-tuning? Besides, do diffusion models require additional fine-tuning to effectively cooperate with the LLM-infused Diffuser? It would be better to see results or insights into integrating the LLM-infused Diffuser into other diffusion models.

---

> ### Author Rebuttal · Authors · 2024-08-06
>
> Dear reviewer nE1N,
>
> Thanks for your comments. We will address your concerns below.
>
> ## Q1: The training and inference cost.
>
> We train the LI-DiT-10B model on a GPU cluster with 1024 NVIDIA A100-80G. The training framework is implemented with Pytorch. We use the gradient checkpointing, mixed precision training and Fully Sharded Data Parallel (FSDP) to enable the large-scale model optimization, and adopt the flash-attention 2 operator for high-efficient computication. The training cost of LI-DiT-10B is about 47500 GPU days, between DALL-E 2[1] (41667 GPU days)  and RAPHAEL[2] (60000 GPU days). Recent works including Stable Diffusion 3[4] and DALL-E 3[5] do not provide the training cost.
>
> We provide the inference cost on both Nvidia A800 GPU and Nvidia H800 GPU in BFLoat16 precision when using both Pytorch and TensorRT to generate images at $1024\times1024$ resolution in 50 steps. The inference time is an average of generating 100 images.
>
> | GPU             |      Pytorch      |     TensorRT      |
> | :-------------- | :---------------: | :---------------: |
> | Nvidia A800 80G | 33.5s / per image | 27.0s / per image |
> | Nvidia H800 80G | 17.1s / per image | 11.2s / per image |
>
> ## Q2: Integrating LLM-infused Diffuser into other diffusion models.
>
> LLM-infused Diffuser provides an effective paradigm to train diffusion basemodel with powerful prompt comprehension capabilities, which can be compatible with different architectures like transformers and U-Nets. The LLM-infused diffuser is jointly trained with the denoising network, serving as a part of the diffusion model. It can not be directly used in other diffusion models. We conduct a supplementary experiment on U-Net based diffusion model with 1B parameter similar to SDXL [6]. The training data and training setting follow that in Figure 2 and Section 4.1. We can observe the significant improvement in unet-based architecture, which further verifies its compatibility.
>
> | Text Encoder                | Denoise Network | LLaMA3-8B | LLaMA3-8B-infused diffuser |
> | --------------------------- | :-------------: | :-------: | :------------------------: |
> | T2I CompBench Average Score |      U-Net      |   37.24   |           50.86            |
>
> If we hope to introduce LM-infused diffuser into existing models like SDXL, it requires extra adapter module and fine-tuning to align LLMs with SDXL. We will release a LLM-infused diffuser with SDXL in the final version.
>
> - [1] Hierarchical Text-Conditional Image Generation with CLIP Latents
> - [2] RAPHAEL: Text-to-Image Generation via Large Mixture of Diffusion Paths
> - [3] PixArt-$\alpha$: Fast Training of Diffusion Transformer for Photorealistic Text-to-Image Synthesis
> - [4] Scaling Rectified Flow Transformers for High-Resolution Image Synthesis
> - [5] Improving Image Generation with Better Captions
> - [6] SDXL: Improving Latent Diffusion Models for High-Resolution Image Synthesis

---

> > ### Comment · Reviewer_nE1N · 2024-08-11
> >
> > Thank the authors for the response, I will keep my score.

---

> > > ### Author Response · Authors · 2024-08-11
> > > **Thanks for your comments**
> > >
> > > We sincerely thank the reviewer for the constructive feedback and support.

---

### Official Review · Reviewer_NTY6 · 2024-07-11

**Soundness:** 3
**Presentation:** 2
**Contribution:** 2
**Rating:** 4
**Confidence:** 4

**Summary:**

This paper presents an investigation into the integration of Large Language Models (LLMs) into text-to-image diffusion models. It identifies issues with using LLMs as prompt encoders, namely misalignment between next-token prediction training in LLMs and the need for discriminative prompt features in diffusion models, as well as positional bias introduced by the decoder-only architecture. The authors propose a new framework to overcome these challenges and introduce an LLM-Infused Diffusion Transformer (LI-DiT) to leverage LLMs effectively in image generation tasks. The paper also discusses broader societal impacts and adheres to ethical guidelines.

**Strengths:**

- The proposed method addresses a significant gap in utilizing LLMs for prompt encoding in diffusion models, offering a new solution to enhance text-to-image generation.
- The paper includes a discussion on potential societal impacts, considering both positive and negative outcomes,

**Weaknesses:**

- The visual results of I2T may exhibit cherry-picking, including Figure 1 and Figure 7. The authors need to provide further explanations. Although they claim there are more prompt-generated results in the appendix, this clearly does not mitigate the risk of cherry-picking.
- Integrating LLMs into diffusion models is not uncommon, and the authors lack discussion and analysis of such works, e.g., [1][2][3]. More in-depth exploration by the authors is needed.
- Honestly, in the visual results of Figure 7, I did not notice any particular advantages of the proposed method.
- Besides the visual effects, the proposed method does not seem to offer significant improvements over existing methods and lacks theoretical support. There is a need for further improvement in novelty.




[1] LLM4GEN: Leveraging Semantic Representation of LLMs for Text-to-Image Generation

[2] UNIMO-G: Unified Image Generation through Multimodal Conditional Diffusion

[3] SUR-adapter: Enhancing Text-to-Image Pre-trained Diffusion Models with Large Language Models

**Questions:**

see weakness

**Limitations:**

Yes

---

> ### Author Rebuttal · Authors · 2024-08-06
>
> Dear reviewer NTY6,
>
> Thanks for your comments. We will address your concerns below.
>
> ## Q1: The contribution and novelty of our work.
>
> Please refer to the Q1 of our global response.
>
> ## Q2: Comparing with other methods adopting LLMs.
>
> Please refer to the Q2 of our global response.
>
> ## Q3: The risk of cherry-pick in visualization.
>
> As stated in the paper, the images in Figure 1 and Figure 7 are randomly sampled. Apart from visualization showcases, the human preference evaluation in Figure 6 also demonstrates the powerful image generation quality and prompt following capability of LI-DiT-10B. Considering the requirements of double-blind review, we will provide the link to our online platform in the final version.
>
> ## Q4: The visualization comparision in Figure 7.
>
> - The image from LI-DiT-10B follows the *tea leaf* in the prompt but it is the lotus leaf for other models. The image from LI-DiT-10B possesses remarkable aesthetic qualities in terms of light and shadow details, whereas the colors in other images appear more muted.
> - The image from LI-DiT-10B better captures the *terrifying atmosphere* described in the prompt.
> - These models can all follow prompts to generate high-quality images.
> - The image from LI-DiT-10B maintains the original form of the crab, whereas crabs in the other images have human body parts. Meanwhile, The image from LI-DiT-10B follows the *red tie* in the prompt.

---

### Official Review · Reviewer_yHpu · 2024-07-11

**Soundness:** 3
**Presentation:** 2
**Contribution:** 3
**Rating:** 6
**Confidence:** 4

**Summary:**

In the context of text-to-image (T2I) generation, this work addresses the problem of leveraging representations from state-of-the-art decoder-based LLMs for conditioning image generation. The authors highlight challenges of leveraging existing LLMs, namely - misalignment in representations due to differing training objectives (autoregressive v/s non-autoregressive) and the effects of positional bias. The authors then propose ways to mitigate this misalignment via prompt changes (leveraging instruction tuning of LLMs), embedding ensembling techniques (for positional bias), and a new DiT based on this ensembled conditioning mechanism. The proposed observations and method show consistent qualitative and quantitative improvements on benchmarks across the board.

**Strengths:**

This paper addresses an important and relevant problem of aligning a) very capable LLM models with b) capable image generation models and understanding why and if there could be a mis-alignment between these two objectives. I like the motivation of leveraging instruction tuned language models and it is understandable why one might want to switch to, or at least explore these models for conditioning text to image generation. Further, the results are impressive and I appreciate the ablation studies conducted for different components proposed in the work, and the custom benchmark to probe the alignment. The paper is presented in a clear and concise manner with appropriate visuals where necessary.

**Weaknesses:**

While the paper shows significant empirical improvements, I have some concerns and questions where it would be great to have a discussion with the authors:

1. Image generation with diffusion models is non autoregressive, while decoder-LLM based representations are trained with auto-regressive models. There is much needed discussion surrounding the choice of models (Encoder decoder, encoder only, decoder only) which does not seem to be addressed in this work. For example, it seems that encoder-only representations are a natural choice for this task (such as InstructOR[1] embeddings, which are tuned for multi task learning in an encoder-only setup, or others from the MTEB [2] benchmark). How do they, or rather would they perform? What is the intuition?


2. The motivation behind adding ensembling, connections with positional bias, and the linguistic token refiner from different LLMs seems lacking. What is intuition here? Why does one need “..The image prompt with instructions to be encoded by multiple frozen LLMs separately”?

3. Re. the positional bias: Given that the LLMs tested in the paper are still small (~2-7B), it may be possible that their “long context” understanding may be limited as an artifact of their instruction tuning. Thus, is the positional bias inherent to these decoder based LLMs or a capability drawback of the smaller LLM on long context itself? That is, can this gap be closed if one employs an LLM with impressive and uniform long context capabilities?

[1] One Embedder, Any Task: Instruction-Finetuned Text Embeddings Su, Hongjin, et al.

[2] Muennighoff, Niklas, et al. "MTEB: Massive text embedding benchmark."

Suggestions:
1. I would encourage the authors to further expand on related work, especially focussing on the alignment between language model types and the task of (non-autoregressive) image generation.
2. (Presentation) The coherence between claims can be improved (it seems that many observations -> design changes) are grouped together.

**Questions:**

(Besides the ones in the above section) In the experiments, what is the average prompt context length (min, max, avg)?

**Limitations:**

Discussed.

---

> ### Author Rebuttal · Authors · 2024-08-06
>
> Dear reviewer yHpu,
>
> Thanks for your comments. We will address your concerns below.
>
> ## Q1: Analyses on the choice of text encoders.
>
> Analyzing the choice of text encoders from encoder-only model, decoder-only model, and encoder-decoder model is one of the core contributions of our paper. In the paragraph starting from Line 41, we analyze the difference in model architecture, optimization target, and performance in the image generation task between T5-like encoder-decoder models and GPT-like decoder-only models. In Section 2.1, we also exploring the ability of different models to retain prompt information. Based on these analyses, we identified the drawbacks of decoder-only LLMs compared to the T5 encoder and designed a series of methods to address these issues.
>
> As expressed in the paragraph starting from Line 41, our work acknowledges the advantages of encoder-only models in terms of model architecture and optimization targets. However, the current paradigm for LLMs is the decoder-only architecture. For example, the top models in the MTEB benchmark like bge-en-icl[1] and stella_en_1.5B_v5 are based on decoder-only LLMs. Encoder-only models like the InstructOR and Sentence T5-XXL[2] are limited by the capabilities of the foundation T5 model, making it difficult to achieve leading performance. We conduct experiments following in Figure 2, simply replacing the text encoder with InstructOR-XL, a fine-tuned T5-XL encoder. The average performance on the T2I-CompBench is 44.06%, similar to the performance of the T5-XL encoder with 43.47%.
>
> The design of the LLM-infused diffuser incorporates the advantages of the encoder-only architecture and aims to leverage advancements in decoder-only LLMs to further enhance the text understanding capabilities of diffusion models.
>
> ## Q2: Ensembling multiple LLMs.
>
> We observe that different LLMs exhibit preferences for prompt encoding, and integrating multiple LLMs can effectively enhance the model's ability to understand prompts as shown in Table 6.  Ensembling multiple encoders has been widely adopted in advanced works including SDXL[3], SD3[4], and recent popular FLUX.1. This indicates that effectively leveraging the capabilities of multiple text encoders is a crucial factor in enhancing the prompt understanding ability of diffusion models.
>
> ## Q3: Using LLMs with long context capabilities.
>
> The prompt lengths used in positional bias evaluation and user input are significantly shorter than the context length of LLMs. For example, the LLaMA3-8B supports the context size of 8192 tokens. However, the prompt lengths in existing test sets and user scenarios are significantly shorter. In the following table, we use the LLaMA3-8B tokenizer to analyze the prompts of different datasets.
>
> | Data                                      | Min  | Max  | Avg  |
> | ----------------------------------------- | :--: | :--: | :--: |
> | T2I-CompBench                             |  2   |  35  |  10  |
> | DPG-Bench                                 |  40  | 175  |  82  |
> | GenEval                                   |  5   |  13  |  8   |
> | Positional Bias Evaluation Benchmark      |  16  |  25  |  19  |
> | Human Evaluation Benchmark                |  5   | 153  |  88  |
> | Sampled 1000 user prompts from Midjourney |  1   | 721  |  85  |
>
> - [1] C-Pack: Packaged Resources To Advance General Chinese Embedding
> - [2] Sentence-T5: Scalable Sentence Encoders from Pre-trained Text-to-Text Models
> - [3] SDXL: Improving Latent Diffusion Models for High-Resolution Image Synthesis
> - [4] Scaling Rectified Flow Transformers for High-Resolution Image Synthesis

---

> > ### Comment · Reviewer_yHpu · 2024-08-10
> >
> > Thank you for the rebuttal. After reading the rebuttal and other reviews, I have raised my score. Looking forward to seeing this discussion in the final version.

---

> > > ### Author Response · Authors · 2024-08-11
> > > **Thanks for your comments**
> > >
> > > We sincerely thank the reviewer for the kind support of our work. We will incorporate the details into our final version.

---

### Official Review · Reviewer_Nr61 · 2024-07-13

**Soundness:** 3
**Presentation:** 3
**Contribution:** 2
**Rating:** 5
**Confidence:** 3

**Summary:**

This work identifies two main reasons for degraded prompt-following ability in image generation with decoder-only transformers: the misalignment between pretraining objective and diffusion's need of discriminative prompt feature, as well as the intrinsic positional bias for decoder-only transformer. The solution is to enhance text representation capability and remove positional bias. This work also proposes a diffusion architecture conditioned on multiple LLMs. The proposed LLM-Infused Diffusion Transformer (LI-DiT) surpasses previous state-of-the-art open-source models in prompt understanding.

**Strengths:**

* Analysis of the reasons that LLMs not working out-of-the-box is interesting.
* The qualitative and quantitative results demonstrate advantages over prior baselines.
* The ablation studies are solid. The contribution of each contribution is demonstrated.
* The paper is easy to read and understand.

**Weaknesses:**

The main weakness of this work is its lacked novelty.
* The Input Prompt part (Sec 3.1) is prompt engineering.
* The Linguistic Token Refiner (Sec 3.1) can be considered as adding more transformer blocks that are trainable and have full-attention, with the LLM weights frozen.
* The cross-attention blocks in Collaborative Refiner (Sec 3.1) can be considered as self-attention of concatenated features without attending to each other within the same LLM tokens.
* The technical contributions in Sec 3.1 do not exhibit significant novelty.

Furthermore, the comparisons with other works are not clear.
* What the sizes of the circles indicates in Fig. 2 is not clear.
* The model sizes of the baseline models under comparison are not listed in Tab. 1.

Finally, this work ignores an existing line of work that combines LLMs with diffusion models [1,2,3]. These works use LLM to generate intermediate representation generation and then generate images conditioned on the intermediate representation.

[1] LayoutGPT: Compositional Visual Planning and Generation with Large Language Models. Feng, et al. NeurIPS 2023.

[2] LLM-grounded Diffusion: Enhancing Prompt Understanding of Text-to-Image Diffusion Models with Large Language Models. Lian, et al. TMLR 2024.

[3] Grounded Text-to-Image Synthesis with Attention Refocusing. Phung, et al. CVPR 2024.

**Questions:**

The reviewer's questions are primarily from the weaknesses section, specifically:
1. What are the sizes of the circles indicate in Fig. 2?
2. What are the model sizes and the training dataset size of the models in Tab. 1?
3. What is the performance of a model if the collaborative refiner is replaced by the simple linguistic token refiner, with input concatenated in the sequence length dimension?

**Limitations:**

The reviewer did not find unaddressed potential negative societal impact. The unaddressed limitations are described in the weakness section.

---

> ### Author Rebuttal · Authors · 2024-08-06
>
> Dear reviewer Nr61,
>
> Thanks for your comments. We will address your concerns below.
>
> ## Q1: The contribution and novelty of our work.
>
> Please refer to the Q1 of our global response.
>
> ## Q2: Comparing with other methods adopting LLMs.
>
> Please refer to the Q2 of our global response.
>
> ## Q3: The sizes of the circles in Figure 2.
>
> The size of each circle indicates the parameter size of the text encoder model. T5-XL with the fewest parameters is marked with the smallest circle. LLaMA3-8B is marked with a large circle.
>
> ## Q4: Model sizes and training dataset sizes in Table 1.
>
> We present the parameters and training dataset sizes of models in Table 1. Some of the models do not provide parameters or training data size in their papers. Considering current works usually use unavailable internal datasets, it is hard to make fair comparisons.  For the effectiveness of our method, please refer to the experiments in the ablation study section.
>
> | Model            | SD v1.5  | SD v2 | SD XL | SD3-1B | DALL-E 2 | PixArt-$\alpha$ | Li-DiT-1B | DALL-E 3 | SD3-8B | Li-DiT-10B |
> | ---------------- | :------: | :---: | :---: | :----: | :------: | :-------------: | :-------: | :------: | :----: | :--------: |
> | Parameters       |   0.9B   | 0.8B  | 2.6B  |   1B   |   6.5B   |      0.6B       |    1B     |    -     |   8B   |    10B     |
> | Image-text Pairs | 2000M[1] |   -   |   -   |   -    | 650M[1]  |     25M[1]      |    30M    |    -     |   -    |   1000M    |
>
> - [1] PixArt-$\alpha$: Fast Training of Diffusion Transformer for Photorealistic Text-to-Image Synthesis
>
> ## Q5: Replace the collaborative refiner with a linguistic token refiner.
>
> We conduct additional experiments to verify using a simple linguistic token refiner for fusing concatenated text embeddings. We can observe that adopting a linguistic token refiner will only bring limited improvement. Meanwhile, when fusing multiple LLMs, the self-attention mechanism in linguistic token refiner requires more computational resources and memory compared to the cross-attention mechanism collaborative refiner.
>
> | Setting | Concat | collaborative refiner | linguistic token refiner |
> | ------- | :----: | :-------------------: | :----------------------: |
> | T2I-avg | 58.32  |       **60.31**       |          58.86           |
> | DPG-avg | 79.04  |       **80.25**       |          79.32           |

---

> > ### Comment · Reviewer_Nr61 · 2024-08-10
> >
> > After reading the rebuttal and other reviews, I still recommend borderline accept. The authors are encouraged to incorporate the discussions about the contributions and the discussion about related works in the general rebuttal section in their work.

---

> > > ### Author Response · Authors · 2024-08-11
> > > **Thanks for your comments**
> > >
> > > We sincerely thank the reviewer for the kind support of our work.  We will incorporate the discussions in the final version.

---

### Author Rebuttal · Authors · 2024-08-06

We sincerely appreciate the valuable time and effort all reviewers have dedicated to review our work. We are pleased to learn that the reviewers generally acknowledge and commend our contributions, including:

- The importance of LLM-infused diffuser in integrating decoder-only LLMs into the diffusion framework. (yHpu, NTY6, nE1N)
- The insightful analysis of the poor performance when adopting decoder-only LLMs as text encoders. (Nr61, yHpu)

- State-of-the-art performance. (Nr61, yHpu, nE1N)
- Solid ablation studies. (Nr61, yHpu).

We express our gratitude for the insightful and constructive suggestions provided by all the reviewers. We also show more high-quality generated images in the attached file. We will provide the link to our online platform in the final version. Here, we address the common concerns raised by the reviewers.

## Q1: The contribution and novelty of our work. (Nr61, NTY6)

The core contribution and novelty of our paper is addressing the issue of poor performance when using powerful decoder-only LLMs as diffusion text encoders. We put more emphasis on investigating the inherent properties of T5-like encoder-decoder models and GPT-like decoder-only models serving as text encoders (paragraph starts with line41), and the reasons behind the poor performance of decoder-only LLMs (Section 2). The simple and effective LLM-infused Diffuser is designed based on novel and contributional observations. Similarly, the key discovery of Imagen[1] is that T5 series models are surprisingly effective, which leads subsequent advanced works to adopt the T5 series models. Our work further advances the application of LLMs within the diffusion framework, enabling the continual development of LLMs to better enhance the text understanding capabilities.

## Q2: Comparing with other methods adopting LLMs. (Nr61, NTY6)

We argue that our approach has significant differences compared to the existing methods that utilize LLMs for prompt encoding.

Current works that utilize LLMs can be categorized into: 1) LLMs first generate the image layout based on the prompt, then the diffusion model completes the image based on this layout[2,3,4] mentioned by reviewer Nr61. 2) Training an extra adapter to align LLM with frozen diffusion models like SD1.5[5] and SDXL[6] for better prompt comprehension capabilities[7,8] mentioned by reviewer NTY6. 3) Leveraging LLMs as text encoders without specific design[9]. UNIMO-G[10] mentioned by reviewer NTY6 adopts MLLMs to take extra images as conditional information, which is not the same task as text-to-image generation.

The contribution of the LLM-infused diffuser does not conflict with the layout approach. The layout methods are usually adopted as the controllable plugin in specific areas like visual composition and number-sensitive tasks. They need to be used in conjunction with a powerful diffusion model. However, the generation quality of each object in the layout still relies on the prompt understanding capability of the diffusion model. When generating a single object with a complex description, the layout approach essentially falls back to directly using the diffusion model for generation. Meanwhile, the layout can only provide the spatial relationship of objects but can not guide the generation of complex object relationships such as *a boy sitting on the shoulder of a man*, while the LLM-infused diffuser can easily deal with it.

The adapter-based methods have not addressed the issues we observed. LLM4GEN[7], which was submitted to arxiv on 30 Jun 2024 after the submission of NeurIPS, also observed that the performance when adopting T5-XL can also easily outperform using larger 13B decoder-only LLMs. However, they did not provide any further analysis and directly used T5-XL as the final text encoder. The following table is the Table 4 in LLM4GEN.

| LLMs       |   Color   |   Shape   |  Texture  |
| ---------- | :-------: | :-------: | :-------: |
| SD 1.5     |   37.65   |   35.76   |   41.56   |
| LLaMA2-7B  |   43.21   |   40.12   |   48.91   |
| LLaMA2-13B |   43.98   |   41.03   |   49.21   |
| T5-XL-1.2B | **45.34** | **43.28** | **51.52** |

Meanwhile, LI-DiT-1B also shows outstanding performance compared with LLM4GEN with fewer parameters.

| Model        | Parameter |   Color   |   Shape   |  Texture  |  Spatial  |
| ------------ | :-------: | :-------: | :-------: | :-------: | :-------: |
| LLM4GEN SDXL |   2.6B    |   73.29   |   57.34   |   67.86   |   22.59   |
| LI-DiT-1B    |    1B     | **74.08** | **59.34** | **69.59** | **27.57** |

Lumina-T2X directly introduces decoder-only LLMs into diffusion transformers, which is similar to our baseline setting.

Compared with the related works above, our method specifically identifies the issues and provides effective solutions. We will include these analyses in the final version.

- [1] Photorealistic Text-to-Image Diffusion Models with Deep Language Understanding
- [2] LayoutGPT: Compositional Visual Planning and Generation with Large Language Models.
- [3] LLM-grounded Diffusion: Enhancing Prompt Understanding of Text-to-Image Diffusion Models with Large Language Models
- [4] Grounded Text-to-Image Synthesis with Attention Refocusing.
- [5] High-Resolution Image Synthesis with Latent Diffusion Models.
- [6] SDXL: Improving Latent Diffusion Models for High-Resolution Image Synthesis
- [7] LLM4GEN: Leveraging Semantic Representation of LLMs for Text-to-Image Generation
- [8] SUR-adapter: Enhancing Text-to-Image Pre-trained Diffusion Models with Large Language Models
- [9] Lumina-T2X: Transforming Text into Any Modality, Resolution, and Duration via Flow-based Large Diffusion Transformers
- [10] UNIMO-G: Unified Image Generation through Multimodal Conditional Diffusion

---

### Public Comment · ~Zecheng_Tang1 · 2025-04-02
**What's the connection between the cases shown in Figure 4 and the textual hidden state we actually need for DiT?**

Hi, thanks for the great work. I notice the cases shown in Figure 4, where the causal LLM generates the ongoing text while T5 repeats the text.

Denote the text as $X$, when we use LLM as a textual information extractor, we usually directly treat the hidden state of $X$ from the last layer of LLM (denote as $f_\theta (X)$) as the representation, and feed  $f_\theta (X)$ to DiT model for image generation.

I have a question, the case in Figure 4 shows the continual generation process of LLM (given $X$ -> predict $Y$) rather than the feature of the representation  $f_\theta (X)$. We just need to use $f_\theta (X)$ for DiT rather than  $f_\theta (Y)$. What's the connection between the cases shown in Figure 4 and the hidden state we actually need?

I hope the authors can answer this question, many thanks!

---

> ### Public Comment · ~Bingqi_Ma1 · 2025-04-02
> **Answer to the question**
>
> Thank you for your interest in our work.
> The purpose of analyzing the content generated by LLMs is to understand the information embedded in hidden states.
> The generated output reflects how the attention mechanism focuses on different aspects of the input prompt.
> With carefully designed system prompts, LLMs can accurately output attributes critical for image generation, such as color and texture.
> This indicates that the features in the hidden states are sensitive to such information and receive higher attention, making them more conducive to the learning process of the diffusion model.

---

> > ### Public Comment · ~Zecheng_Tang1 · 2025-04-03
> > **Response to authors**
> >
> > Dear authors, thanks for your response. Now I understand.

---

### Decision · Program_Chairs · 2024-09-25

**Decision:**

Accept (poster)

**Comment:**

This paper explores the role of LLMs in prompt encoding for diffusion models, especially in terms of the learned representation and effect of positional bias. The paper received 2 weak accepts, 1 borderline reject, and 1 borderline accept recommendations from reviewers. Positive points included interesting analysis, clear writing, good results, and extensive ablation studies. Negative points included limited novelty, some missing related work, some unclear comparisons, and lacking discussion on training and inference costs and scalability. Most of these concerns were addressed by the rebuttal. The reviewer who gave the borderline reject rating did not participate in the post-rebuttal discussions; however, the ACs felt that the reviewer’s major concerns were addressed by the rebuttal. Overall, after carefully considering the paper, rebuttal, and discussions, the ACs feel that the paper makes a positive contribution and recommend accept. Please incorporate the rebuttal points into the camera ready version.